# Cellulose Acetate-Supported Copper as an Efficient Sustainable Heterogenous Catalyst for Azide-Alkyne Cycloaddition Click Reactions in Water

**DOI:** 10.3390/ijms24119301

**Published:** 2023-05-26

**Authors:** Salah-Eddine Stiriba, Lahoucine Bahsis, Elhouceine Benhadria, Khaoula Oudghiri, Moha Taourirte, Miguel Julve

**Affiliations:** 1Instituto de Ciencia Molecular/ICMol, Universidad de Valencia, C/Catedrático José Beltrán 2, 46980 Paterna, Valencia, Spain; miguel.julve@uv.es; 2Laboratoire de Chimie Analytique et Moléculaire (LCAM), Faculté Polydisciplinaire de Safi, Université Cadi Ayyad, Safi 46030, Morocco; bahsis.lahoucine@gmail.com; 3Département de Chimie, Faculté des Sciences d’El Jadida, Université Chouaïb Doukkali, El Jadida 24000, Morocco; benhadria.smc@gmail.com; 4Laboratoire de Recherche en Développement Durable et Santé, Faculté des Sciences et Techniques de Marrakech, Université Cadi Ayyad, Marrakech 40000, Morocco; khaoulaoudghiri08@gmail.com (K.O.); m.taourirte@uca.ma (M.T.)

**Keywords:** cellulose acetate, biopolymers, catalyst immobilization, CuAAC, 1,2,3-triazoles

## Abstract

A new sustainable heterogeneous catalyst for copper-catalyzed azide-alkyne cycloaddition reaction (CuAAC) was investigated. The preparation of the sustainable catalyst was carried out through the complexation reaction between the polysaccharide cellulose acetate backbone (CA) and copper(II) ions. The resulting complex [Cu(II)-CA] was fully characterized by using different spectroscopic methods such as Fourier-transform infrared spectroscopy (FTIR), Scanning Electron Microscopy (SEM), Energy Dispersive X-ray (EDX), Ultraviolet-visible (UV-vis), and Inductively Coupled Plasma (ICP) analyses. The Cu(II)-CA complex exhibits high activity in the CuAAC reaction for substituted alkynes and organic azides, leading to a selective synthesis of the corresponding 1,4-isomer 1,2,3-triazoles in water as a solvent and working at room temperature. It is worth noting that this catalyst has several advantages from the sustainable chemistry point of view including no use of additives, biopolymer support, reactions carried out in water at room temperature, and easy recovery of the catalyst. These characteristics make it a potential candidate not only for the CuAAC reaction but also for other catalytic organic reactions.

## 1. Introduction

Recently, the preparation, characterization, and catalytic activity of heterogeneously catalyzed reactions have been reviewed focusing on their positive and negative points as well as on using bio-heterogeneous catalytic systems that involve biological macromolecules, such as cellulose, alginate, chitosan, etc. as bio-supports for metal catalysts [1,2]. The great interest in sustainable chemistry has led to the investigation of natural resources such as biopolymers. Among the most abundant naturally occurring and biodegradable polymers, cellulose known for its large number of interesting structure-related properties was utilized in various industrial applications, specifically in water treatment, cosmetics, and the industry of paper [3,4,5]. The introduction of functional groups on the cellulose structure has attracted great attention due to the possibility of modification of the chemical and physical properties of the cellulose surface [6,7,8], and their ability to catch metal ions is one of the best methods to introduce the eco-friendly character of the heterogeneous catalysts [9,10,11,12]. Since its introduction by Sharpless, the click chemistry concept is reported to feature high regioselectivity, high yields, and only low reaction times for the synthesis of a great variety of organic molecules with a potential application [13].

The copper-catalyzed azide-alkyne cycloaddition reaction (CuAAC) is the most popular reaction in the click chemistry regime. It is an excellent ligation process between azides and alkynes for the selective synthesis of 1,4-disubstituted-1,2,3-triazoles [14,15]. Triazole derivatives are excellent candidates in medicinal chemistry [16], biological science [17], and material chemistry [18]. The development of new heterogeneous catalytic systems respecting all reported green chemistry principles has experienced a significant increase in the last decade due to their easy recovery and high reusability [19,20,21,22]. Heterogeneous catalysis with metal ions on bio-supports is one of the most developed and efficient methods with an eco-friendly character [23,24,25,26,27]. In this regard, the development of new catalysts to afford 1,2,3-triazole moiety with high yields under sustainable conditions has attracted a lot of attention in the last decades [28,29,30]. We report here the heterogenization of copper(II) ions on cellulose acetate (CA) via the coordination of copper(II) by carbonyl and other oxygen-containing groups on the CA surface. The obtained catalyst (Cu(II)-CA) is highly active and regioselective in the synthesis of the corresponding 1,4-disubstituted-1,2,3-triazoles at room temperature using water as solvent. The heterogeneity and reusability of the prepared sustainable catalyst have also been investigated.

## 2. Results and Discussion

The immobilization of copper(II) ions on the cellulose acetate (CA) surface was conducted via a simple complexation process by using copper(II) chloride dihydrate as a source of copper(II) and water as solvent at room temperature. In fact, this immobilization reaction occurs through a complexation reaction between Cu(II) ions and both carbonyl oxygen and hydroxyl groups as donors on the CA surface (Figure 1). The Inductively Coupled Plasma (ICP) analysis was performed to determine the contents of copper(II) ions in the Cu(II)-CA catalyst which was found to be 1.37% *w*/*w*. The morphology and structural investigations of the prepared Cu(II)-CA catalyst were carried out by means of Fourier-transform infrared spectroscopy (FTIR), UV-Vis, Scanning Electronic Microscopy (SEM), and Energy dispersive X-ray (EDX) spectroscopy.

### 2.1. Characterization of the Cu(II)-Catalyst

Cellulose acetate (CA) before and after its complexation reaction with copper(II) ions was analyzed by FTIR in order to gain more insights into the chemical structure of the prepared catalyst. The obtained results are summarized in Figure 1. The FTIR spectrum of the pure CA in the high-frequency region shows a broad band centered at ca. 3500 cm^−1^ which is related to OH groups plus other absorption peaks at 2948 and 2880 cm^−1^ due to C-H asymmetric stretching vibrations. The peaks at 1733 and 1431 cm^−1^ are assigned to the C=O stretching and C-O-H in-plane bending at C6, respectively. Other absorptions around 1366 cm^−1^ are attributed to C-O-H bending at C2 or C3. C=O bending vibration was observed in the FTIR spectrum at 1029 cm^−1^. Finally, the absorption peaks at 1215 and 902 cm^−1^ are attributed to C-O-C stretching of the β-(1-4) glycosidic linkage, which is the characteristic link in the cellulose structure. As far as the infrared spectrum of Cu(II)-CA is concerned, no new bands appeared, the only change being the intensity of all characteristic bands of CA (see Figure 1), suggesting that the possible interactions of functional ester and hydroxyl groups from CA with copper(II) ions in the Cu(II)-CA catalyst are weak, and also keeping in mind that the copper contents therein is very small.

The morphology and surface analysis of the obtained Cu(II)-CA catalyst was performed through SEM and EDX techniques (see Figure 2). The SEM image for the pure cellulose acetate shows no homogeneous surface with the presence of a microporous pattern, indicating the ability of this material to adsorb metal ions (Figure 2a) [31,32,33]. In the case of the Cu(II)-CA catalyst, the SEM image shows a low dispersity of copper (Figure 2b) and the absence of the metal aggregate which is due to the high solubility of copper(II) chloride in water, reducing the interaction of copper(II) ions with the functional groups on the cellulose acetate surface. The amount of copper on the surface of the polysaccharide cellulose acetate was examined by EDX analysis. The obtained result confirms the presence of copper in the Cu-CA catalyst and the copper loading of 1.87 wt% (Figure 2c). The difference between the obtained copper loading by ICP and EDX analyses is that EDX analysis can measure only the copper which is present on the surface of the cellulose acetate but it is unable to detect the copper loading inside of this polysaccharide polymer.

The optical characteristics of the Cu(II)-CA catalyst were also investigated by UV-Vis spectroscopy in the solid state. As shown in Figure 3, no absorption was located in the case of the pure cellulose acetate between 200 to 1000 nm. Meanwhile, the UV-Vis analysis of the Cu(II)-CA catalyst shows the occurrence of two new bands. The first one which is located in the UV domain at around 290 nm (Figure 3) is attributed to the presence of the Cu^2+^–O^−^ charge-transfer transitions [34,35,36,37]. The other one is a broad band located around 800 nm, and it is attributed to d-d transitions of copper(II) ions immobilized on the cellulose acetate polysaccharide.

### 2.2. Catalytic Tests

The prepared material, Cu(II)-CA, was then tested in CuAAC reactions under strict click reaction conditions. As a model reaction, the one between phenylacetylene (**1a**) and benzyl azide (**2a**) was selected for a systematic evaluation under various conditions (Figure 2).

The experimental results show that the use of CuCl_2_ only affords a moderate yield of the desired product (**3a**) after 24 h at room temperature (Table 1). Importantly, the Cu(II)-CA catalyst leads to selective synthesis of one regioisomer triazole derivative, 1,4-disubstituted 1,2,3-triazole, in an excellent yield (99%) at room temperature using water as solvent. Subsequently, the effect of the amount of the catalyst on the efficiency of the catalyzed CuAAC reaction was also investigated (Table 1). The use of 5 mol% of Cu(II)-CA leads to an excellent yield after only 8 h, and the decrease in catalyst loading is not good for CuAAC reactions—see entries 7–9. However, no significant promotion in the yield was observed increasing the amount of catalyst to 20 mol%. Additionally, when the reaction temperature was increased to 60 °C, the results showed that this Cu(II)-CA catalyst led to an excellent yield (>90%) within 4 h (Table 1, entries 10–12). The long reaction time (~8 h) in the CuAAC reaction using Cu(II)-CA catalyst is due to the reaction rate for the formation of copper(I), the catalytic species for the CuAAC reaction, which is generated by the reduction of Cu(II) by terminal alkyne via the oxidative alkyne homocoupling reaction [38].

The ligation of various alkyne and azide derivatives was then investigated under the optimized CuAAC reaction conditions by using the Cu(II)-CA catalyst (Table 2). In all cases, the electron-donating, electron-withdrawing, or heterocycle substituents on the studied azides and terminal alkynes have no significant effects when using this catalyst. In fact, the final 1,2,3-triazole derivatives were obtained in excellent yields without further purification by conventional methods as confirmed by their analysis through ^1^H and ^13^C NMR spectroscopy (see Appendix A).

### 2.3. Reusability of Cu(II)-CA Catalyst

To examine the recyclability and stability of the prepared catalyst in the CuAAC reaction, a model reaction was chosen between phenylacetylene (**1a**) and benzyl azide (**2a**) under the optimized reaction conditions (Figure 3). The results showed that a moderate yield was obtained after three cycles (62%) (Table 3). The morphology of the reused catalyst was examined by SEM and EDX analyses after four cycles (Figure 4). The results show that the morphology of the fresh and reused Cu(II)-CA catalyst are almost similar. Moreover, the copper percentage in the reused catalyst was also investigated by EDX analysis and the results show a low amount of copper on the surface of the recovered Cu(II)-CA catalyst which can explain the low yield achieved after four cycles. The reusability, catalytic activity, and biocompatibility of this Cu(II)-CA catalyst make it a potential candidate not only for CuAAC reactions but also for other copper-catalyzed organic reactions.

### 2.4. Heterogeneity Test

The heterogeneity test of the prepared catalyst for the CuAAC reaction was also investigated through a hot filtration test (Figure 5). The cycloaddition reaction between benzyl azide and phenylacetylene in the presence of the prepared catalyst was performed in a reaction tube at 60 °C using water as solvent. At reaction halftime, the CuAAC reaction was stopped, and then, the catalyst was removed by hot filtration. The reaction filtrate was then stirred at the same reaction temperature for a further reaction time of 2 h. The results confirm that no corresponding 1,2,3-triazole was obtained after the hot filtration of the catalyst, unveiling the heterogeneous nature of the Cu(II)-CA catalyst.

### 2.5. Mechanistic Studies

The reaction mechanism for the CuAAC reaction catalyzed by Cu(II)-CA is reported in Figure 4. Based on the previously reported mechanism of CuAAC [39,40,41,42,43,44,45], the first step is the formation of the catalytic copper(I) species by the reduction of the copper(II) ions in the presence of terminal alkyne via the homocoupling process [46]. The coordination of the obtained copper(I) species with alkynes conducts to the formation of the dinuclear acetylide-copper complex. This dinuclear copper(I) species reacts with the organic azides, resulting in a six-membered ring formation. The rearrangements of this intermediate lead to the formation of the 1,4-disubstituted 1,2,3-triazole and subsequent regeneration of the Cu(II)-CA complex in the presence of dioxygen (Figure 4).

### 2.6. Comparison with Other Catalytic Methods

To illustrate the merits of the Cu(II)-CA catalyst in the CuAAC reaction, its catalytic activity was compared with other reported catalytic systems. The CuAAC reaction between phenylacetylene (**1a**) and benzyl azide (**2a**) was chosen as the model reaction to gain more light on the efficiency of our catalytic system. As shown in Table 4, the Cu(II)-CA catalyst is similar to the Cu(II) ions complexed with the naturally occurring biopolymers as well as with the modified biopolymer. Excellent results were observed compared to the Cu(II)-polyethyleneimine both in the reactivity and reaction time. Moreover, the formation of copper(I), which is the catalytic species for the CuAAC reaction, does not require an external reducing reagent in our case. All these qualities confirm that this catalyst is a good eco-friendly candidate for other organic synthesis reactions catalyzed by Cu(II)/Cu(I) ions.

## 3. Materials and Methods

### 3.1. General Experimental Information

All used reagents in this investigation were purchased from Sigma-Aldrich. The thin layer chromatography (TLC) plate (Merck Kieselgel 60 F254, Darmstadt, Germany) was used to monitor the catalytic reactions. All obtained products were characterized through ^1^H and ^13^C NMR analysis by using the BRUKER DRX-300 AVANCE spectrometer (University of Valencia, Valencia, Spain) and CDCl_3_, DMSO, and MeOD as solvents. FT-IR spectra were taken on a Nicolet spectrophotometer 5700. The scanning electron microscopy (SEM) images were obtained by means of an electron microscopy Philips XL-30 ESEM coupled to Tescan Vega-3 w/EDX (University of Valencia, Valencia, Spain).

### 3.2. Preparation of the Cu(II)-Catalyst

The complexation process between cellulose acetate and copper(II) ions was achieved by the addition of cellulose acetate (1 g) to an aqueous solution of copper(II) chloride dihydrate ([CuCl_2_] = 0.093 mol/L in 10 mL of water). The resulting mixture was stirred overnight at room temperature, and the obtained biomaterial was filtered off, washed with water, and then dried overnight. The copper(II)-containing cellulose acetate was characterized by FT-IR, SEM-EDX, UV-Vis, and ICP spectroscopy.

### 3.3. Catalytic Synthesis of 1,2,3-Triazole Derivatives

The corresponding azides (0.6 mmol) and alkyne (0.5 mmol) derivatives plus 3 mol% of Cu(II)-CA catalyst were added to 3 mL of water under continuous magnetic stirring at room temperature, and the reaction was monitored by TLC. After the reaction completion, the resulting mixture was then diluted by adding ethyl acetate. The catalyst was recovered by simple filtration and then washed, dried, and stored for the next cycle. The solvent of the organic phase was then removed under vacuum to afford the pure corresponding 1,2,3-triazole derivatives (**3a–i**).

## 4. Conclusions

In conclusion, the preparation and the characterization via the immobilization of copper(II) ions on the cellulose acetate (CA) (Cu(II)-CA) were investigated. The Cu(II)-CA catalyst was characterized by FT-IR, UV-Vis, SEM, EDX, and ICP analyses. Such a catalyst exhibits high catalytic activity and selectivity for the synthesis of 1,4-disubstituted-1,2,3-triazoles via CuAAC reaction using water as solvent at room temperature. The stability and heterogeneity of this catalyst were explained by the coordination of carbonyl and hydroxyl groups of the cellulose acetate with copper(II) ions. The simple separation of 1,2,3-triazoles, the reusability and heterogeneity of this catalyst, the various substrate scope, and the use of water as a reaction medium make this catalyst sustainable for CuAAC reactions. In light of these features, one can anticipate that this Cu(II)-CA material will be a very useful catalytic system for other copper-mediated organic reactions under mild eco-friendly reaction conditions.

## Data Availability

Not applicable.

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
