# Peer review of "Cellulose Acetate-Supported Copper as an Efficient Sustainable Heterogenous Catalyst for Azide-Alkyne Cycloaddition Click Reactions in Water"

_ijms, 2023, doi:10.3390/ijms24119301_

Round 1

Reviewer 1 Report

The work reported in this manuscript is interesting and well presented. However, there should be further improvement and revision before the acceptance. Some comments are:

1.     Image quality needs to be improved,for example Figure 1 e and f.

2.     Many obvious writing errors, please revise carefully.

3.     The discussion is poor, please improve, the reaction mechanism should be supplemented.

4.     In scheme 1, Is there direct experimental evidence for whether the hydroxyl groups on cellulose acetate can remove protons and coordinate with copper ions?

5.     Some articles must be cited in the paperï¼›Journal of Molecular Structure, 2023, 1286, 135580; Fuel, 2023, 342, 127890; Mol. Catal. 535 (2023), 112895; Applied Surface Science, 2022, 589,153002.

Many obvious writing errors, please revise carefully

Author Response

Point-by-point responses to the reviewers:

Reviewer #1:

The work reported in this manuscript is interesting and well presented. However, there should be ‎further improvement and revision before the acceptance. Some comments are:‎

1. Image quality needs to be improved,for example Figure 1 e and f.

Authors: We thank the reviewer for drawing our attention to this point. The image quality of all Figures were improved in the revised version.

2. Many obvious writing errors, please revise carefully.

Authors: Spelling and style were checked throughout the main text of the manuscript.

3. The discussion is poor, please improve, the reaction mechanism should be supplemented.

Authors: Additional discussion details were added in the characterization part and the reaction mechanism of the studied process was provided.

4. In scheme 1, Is there direct experimental evidence for whether the hydroxyl groups on cellulose acetate can remove protons and coordinate with copper ions?

Authors: The chemical bond between copper(II) ions and alcoholate in the processs of immobilization of copper(II) ions on the cellulose backabone is supported by the presence of a new absorbane peak assigned to the Cu—O in the UV-vis spectrum as pointed out in previous reports [refs 34-37].

5. Some articles must be cited in the paper;Journal of Molecular Structure, 2023, 1286, 135580; Fuel, 2023, 342, 127890; Mol. Catal. 535 (2023), 112895; Applied Surface Science, 2022, 589,153002.

Authors: We thank the reviewer for drawing our attention to these references. However, the content of these references is out of the scope of this study. In fact, they report studies on CO oxidation under various reaction conditions.

Reviewer 2 Report

Stiriba et al. report the elaboration of a sustainable heterogenous catalyst based on the Cu(II) supported on the polysaccharide cellulose acetate backbone for cycloaddition reaction of alkynes and azides (CuAAC). The obtained catalytic system was characterized by FTIR, SEM, EDX, UV-Vis, and ICP analyses. Before publication, the next issues have to be addressed:

1) Introduction: the part about CuAAC have to be started from a new paragraph.

2) The authors have to point out the benefit of the heterogeneous catalysts for CuAAC vs homogeneous ones. And also cite some fresh papers on the homogeneous and homogeneous catalysts to show actuality and importance of the direction. For example:

Catalysts 2023, 13, 130

Molecules 2022, 27, 16

Applied Organometallic Chemistry 2022, 36, e684

J. Catal. 2020, 390, 37-45

Organometallics 2022, 41, 2154-2169

Nanomaterials 2022, 12, 1070

ACS Appl. Mater. Interfaces 2021, 13, 33091–33101, etc.

3) Give a transcript for the abbreviation “ICP” (page 2, line 64).

4) On Scheme 1, the Cu ion is bonded with hydroxy group forming alcoholate, however the authors still see the free OH group at ca. 3500 cm-1 on FTIR spectra after complexation (Figure 1) that is contradictory to the illustration.

5) No entries 16-21 founded in Table 1 as indicated in the text (page 5, line 131). It is better indicating the index “c” in the column “entry” and not in “yield” (Table 1).

6) Instead of “pgenerated” have to be “generated” (line 133? Page 5).

7) How about to use of Cu(I) salts for the preparation of the cellulose supported catalyst? Probably, the activity can be increased?

8) Line 134. A new paragraph has to be started and probably this text can be moved after Table 1.

9) Why the selectivity can be changed for the CuAAC (as the authors point by the sentence “also a lack of non-change selectivity”, page 5, Line 152). It is no make sense to point out it as for the CuAAC only 1,4-substitued triazoles can be formed.

10) Overall, after the recycling experiments it is not obvious that the Cu(II)-CA works as a heterogeneous catalyst and can be reusable (the 1.5x decrease for 4th cycle). Moreover, it seems that the CuCl2 bounded to the cellulose only by coordination bonds and not covalently as shown in Scheme 1. So, I would not argue about the recyclability and about heterogeneity character in the manuscript as a main message.

11) I’m not sure that the leaching of the Cu(II) ions during the recycling “make this catalyst more competitive for sustainable CuAAC reactions” as the authors mentioned in the conclusion (page 9, line 227). Have to be rewritten.

12) The deuterated DMSO and MeOD was used for the record of NMR spectra, which was not indicated in the general remarks (part 3.1).

13) For ref. 2. no pages and no DOI are given. Moreover, ref. 2 and ref. 19 are similar.

14) SI: for compound 3f (page S15) no integration is provided.

15) The authors state “Moreover, the obtained 1,2,3-triazoles did not require any further purification by conventional methods as confirmed by their analysis by 1H and 13C NMR spectroscopy (see Supporting Information)”. For curiosity, I looked the SI of the similar paper published by the same authors (Catalysts 2022, 12(10), 1244), and found that several spectra seem to be identical (for example, see for compounds 3b vs 3b (in published paper), 3c vs 3i (in published paper)). So, it is a big question that the reaction was clean or the authors just provided the spectra of the pure products from their previous published paper?

The English have to be checked.

Author Response

Point-by-point responses to the reviewers:

Reviewer #2:

Stiriba et al. report the elaboration of a sustainable heterogenous catalyst based on the Cu(II) supported on the polysaccharide cellulose acetate backbone for cycloaddition reaction of alkynes and azides (CuAAC). The obtained catalytic system was characterized by FTIR, SEM, EDX, UV-Vis, and ICP analyses. Before publication, the next issues have to be addressed:

1) Introduction: the part about CuAAC have to be started from a new paragraph.

Authors: As suggested by the reviewer, a new paragraph was started introducing CuAAC as shown on line 48, page 2.

2) The authors have to point out the benefit of the heterogeneous catalysts for CuAAC vs homogeneous ones. And also cite some fresh papers on the homogeneous and homogeneous catalysts to show actuality and importance of the direction. For example:

Catalysts 2023, 13, 130

Molecules 2022, 27, 16

Applied Organometallic Chemistry 2022, 36, e684

  1. Catal. 2020, 390, 37-45

Organometallics 2022, 41, 2154-2169

Nanomaterials 2022, 12, 1070

ACS Appl. Mater. Interfaces 2021, 13, 33091–33101, etc.

Authors: We appreciate the valuable suggestion of these interesting references. The following references: Catalysts 2023, 13, 130 and ACS Appl. Mater. Interfaces 2021, 13, 33091–33101 that report works on heterogeneous CuAAC are in line with the chemistry of this manuscript and therefore, they were cited in the introduction section [refs. 22 and 27].

3) Give a transcript for the abbreviation “ICP” (page 2, line 64).

Authors: The abbreviation of ‘’ICP’’ was added on page 2.

4) On Scheme 1, the Cu ion is bonded with hydroxy group forming alcoholate, however the authors still see the free OH group at ca. 3500 cm-1 on FTIR spectra after complexation (Figure 1) that is contradictory to the illustration.

Authors: The free OH group at ca. 3500 cm-1 on FTIR spectra shwon after complexation of Cu(II) with cellulose acetate backbone is due to the water molecules completing the eight-coordinating sphere of the Cu(II) complex.

5) No entries 16-21 founded in Table 1 as indicated in the text (page 5, line 131). It is better indicating the index “c” in the column “entry” and not in “yield” (Table 1).

Authors: The entries 16-21 were modified and the correct entries appear now as 10-12, see page 5. The index “c” was also added in the column “entry”.

6) Instead of “pgenerated” have to be “generated” (line 133? Page 5).

Authors: Spelling and style were checked throughout the main text of the manuscript.

7) How about to use of Cu(I) salts for the preparation of the cellulose supported catalyst? Probably, the activity can be increased?

Authors: Cu(I) salts were not used in this type of protocol due to their thermodynamic unstability in the presence of oxygen atmosphere and disproportion in water as solvent. In this study, the catalytically active copper(I) species is generated via the reduction of Cu(II) by the alkyne derivative into Cu(I), a reaction known as Glaser reaction, see references : 10.1002/jlac.18701540202 ; 10.1039/C4RA02416H.

8) Line 134. A new paragraph has to be started and probably this text can be moved after Table 1.

Authors: As suggested by the reviewer, a new paragraph was started on line 146 and moved after Table 1.

9) Why the selectivity can be changed for the CuAAC (as the authors point by the sentence “also a lack of non-change selectivity”, page 5, Line 152). It is no make sense to point out it as for the CuAAC only 1,4-substitued triazoles can be formed.

Authors: We thank the reviewer for this remark. The mentioned sentence was removed, see page 6.

10) Overall, after the recycling experiments it is not obvious that the Cu(II)-CA works as a heterogeneous catalyst and can be reusable (the 1.5x decrease for 4th cycle). Moreover, it seems that the CuCl2 bounded to the cellulose only by coordination bonds and not covalently as shown in Scheme 1. So, I would not argue about the recyclability and about heterogeneity character in the manuscript as a main message.

Authors: Several arguments point to the fact that Cu(II)-CA is a heterogenous catalysts. Firstly, this catalysts was recovered and reused in several tests, showing a remarkable activity in spite of its decrease of it in the successive runs, whic is most probably due to a leaching process or fatigue. Secondly, the heterogenetity test is an excellent method that show the stability of the heterogenous catalyst at high temperatures and so, its continous activity as heterogenous catalyst. Regarding, the nature of the chemical bond between Cu(II) and the cellulose backbone, we are dealing with a Cu(II)-containing macromolecular complex where the chemical bonds between Cu(II) ions and the atoms of the acetate cellulose backbone are of mixed nature having covalent and ionic conributions, which make this macrocomplex a stable system.

11) I’m not sure that the leaching of the Cu(II) ions during the recycling “make this catalyst more competitive for sustainable CuAAC reactions” as the authors mentioned in the conclusion (page 9, line 227). Have to be rewritten.

Authors: Following the reviewer's suggestion, the reported sentence was revised and now appeares as “make this catalyst sustainable for CuAAC reactions”

12) The deuterated DMSO and MeOD was used for the record of NMR spectra, which was not indicated in the general remarks (part 3.1).

Authors: We thank the reviewer for this remark. The deuterated DMSO and MeOD solvents were mentioned in the general experimental part, see page 9.

13) For ref. 2. no pages and no DOI are given. Moreover, ref. 2 and ref. 19 are similar.

Authors: All references were checked.

14) SI: for compound 3f (page S15) no integration is provided.

Authors: The integration of compound 3f was added in the SI.

15) The authors state “Moreover, the obtained 1,2,3-triazoles did not require any further purification by conventional methods as confirmed by their analysis by 1H and 13C NMR spectroscopy (see Supporting Information)”. For curiosity, I looked the SI of the similar paper published by the same authors (Catalysts 2022, 12(10), 1244), and found that several spectra seem to be identical (for example, see for compounds 3b vs 3b (in published paper), 3c vs 3i (in published paper)). So, it is a big question that the reaction was clean or the authors just provided the spectra of the pure products from their previous published paper?

Authors: All 1,2,3-triazoles reported in this submission were already reported either by us or by other teams. Once obtained under the reported reaction conditions of this submission, they were analyzed by TLC and their melting points were compared with pure product and subsequently analyzed by 1H and 13C NMR. We provided spectra from previous works by us to ensure the putity of the final 1,2,3-triazole products because the 1H and 13C NMR of the new obtained 1,2,3-triazoles show residual protons peaks of solvents such as acetone, water and ethyl acetate. New NMR spectra of the obtained 1,2,3-triazole derivatives are now provided in SI.

Reviewer 3 Report

This manuscript reports the development of new catalyst for copper click chemistry based on cellulose. The catalyst proposed works well in water and authors claim it to be heterogenous. The cellulose materials are characterized with a series of techniques: EDX, IR, SEM and ICP. Results are clearly presented, but some of the results seem to be in contradiction.

This paper might be of interest to researchers interested in green chemistry and copper catalyzed C-C bond forming reactions.

A main concern of the manuscript is that the authors claim that the catalyst is heterogeneous, however the experiment to prove this, “hot filtration”, seems to provide results contradicting other results. For instance, the catalyst quickly degrades after a few cycles (60% yield after 3 cycles) and the content of copper as determined by EDX decreases from 1.87% to 0.02%. Such a decrease indicates that the cellulose-copper bonds are labile and hence a lot of the copper must be in solution during the catalysis. This seems to contradict the results from the “hot filtration” test. Nevertheless, this is an interesting study, and the manuscript is well written and presented. This reviewer believes that the readers of Int J Mol Sci will enjoy the content of the manuscript and considers it suitable for publication after the following points have been addressed:

1.      On page two, can the authors add the carbon labelling scheme to the Scheme 1, please? The text describing the IR refers to C2, C3 and C6 and those do not appear in the scheme.

2.      While describing changes in the IR, the authors describe that no new bands appear when comparing the IR of cellulose to the cellulose-Cu catalyst. This is not surprising, giving that the copper content of the catalyst is only 0.7%. It is surprising that they observe such a difference in the intensity of the bands, and it begs to wonder if the difference arises from a lack of an internal reference.

3.      The authors claim that the copper content by EDX is 1.87% wt, while the copper content by ICP is 0.7% wt. This is a large discrepancy that can lead to extremely large differences in catalyst loading. One simple question is if the two samples measured by the different techniques were obtained from different batches. Indicating that the copper absorption by the cellulose is hugely variable. The authors explain the difference between the two values by the different techniques saying that the EDX only gives surface content, if this was correct, then the surface content by EDX should be lower than the total content obtained by ICP. This result is hugely contradictory and it must be addressed prior to publication.

4.      On the UV spectrum the authors claim that the 290 nm band observed is due to the Cu-O bond. Do the authors mean that this is a n-pi* transition from the oxygen n orbital?

5.      On page 5, author make the following statement: “… and decreasing the catalyst loading is not good for CuAAC reactions”. Such statement should be supported with a reference to literature.

6.      Later on the same paragraph the authors refer to Table 1 entries 16-21. Such entries do not exit in the table. The last entry is number 12. Please address this.

7.      The authors claim that they always obtain >90% isolated yields. It is surprising to see that they do not observe any alkyne homocoupling reaction, given that the alkyne is the limiting reagent. Do the authors have any explanation for this observation?

8.      The authors claim in table 2 that the catalyst loading is 3 mol% Cu. To calculate that 3% loading did they use the 0.7% content or did they use the 1.87%? This also has implications later on on Table 3 about the reusability of the catalyst. Given that the copper content after 3 cycles is only 0.02% weight by EDX, this implies that more than 99% of the copper present in the catalyst has been lost after 3 cycles. This presents a clear barrier to the use of this catalyst, since one of the main advantages of heterogeneous catalysts and being able to use them in continues processing, instead of batch chemistry, is rapidly degraded.

9.      What are the reaction conditions for the reusability in Table 3? Are they the same as Table 2? Can the author please include the conditions in the table caption as they did in table 2 please?

Author Response

Point-by-point responses to the reviewers:

Reviewer #3:

This manuscript reports the development of new catalyst for copper click chemistry based on cellulose. The catalyst proposed works well in water and authors claim it to be heterogenous. The cellulose materials are characterized with a series of techniques: EDX, IR, SEM and ICP. Results are clearly presented, but some of the results seem to be in contradiction.

This paper might be of interest to researchers interested in green chemistry and copper catalyzed C-C bond forming reactions.

A main concern of the manuscript is that the authors claim that the catalyst is heterogeneous, however the experiment to prove this, “hot filtration”, seems to provide results contradicting other results. For instance, the catalyst quickly degrades after a few cycles (60% yield after 3 cycles) and the content of copper as determined by EDX decreases from 1.87% to 0.02%. Such a decrease indicates that the cellulose-copper bonds are labile and hence a lot of the copper must be in solution during the catalysis. This seems to contradict the results from the “hot filtration” test. Nevertheless, this is an interesting study, and the manuscript is well written and presented. This reviewer believes that the readers of Int J Mol Sci will enjoy the content of the manuscript and considers it suitable for publication after the following points have been addressed:

1. On page two, can the authors add the carbon labelling scheme to the Scheme 1, please? The text describing the IR refers to C2, C3 and C6 and those do not appear in the scheme.

Authors: Following the reviewer's suggestion, carbon numbering was added in Scheme 1.

2. While describing changes in the IR, the authors describe that no new bands appear when comparing the IR of cellulose to the cellulose-Cu catalyst. This is not surprising, giving that the copper content of the catalyst is only 0.7%. It is surprising that they observe such a difference in the intensity of the bands, and it begs to wonder if the difference arises from a lack of an internal reference.

Authors: We thank the reviewer for drawing our attention to this remark. The internal reference was used in the FTIR analysis, but no new bands were found due to the low loading of copper ions in the Cu(II)-CA catalyst.

3. The authors claim that the copper content by EDX is 1.87% wt, while the copper content by ICP is 0.7% wt. This is a large discrepancy that can lead to extremely large differences in catalyst loading. One simple question is if the two samples measured by the different techniques were obtained from different batches. Indicating that the copper absorption by the cellulose is hugely variable. The authors explain the difference between the two values by the different techniques saying that the EDX only gives surface content, if this was correct, then the surface content by EDX should be lower than the total content obtained by ICP. This result is hugely contradictory and it must be addressed prior to publication.

Authors: The ICP analysis was performed using 100 mg of prepared catalyst while the EDX analysis was performed using only 10 mg of this catalyst. These findings indicate that the large copper content is located in the CA surface, which can explain the high difference values between ICP and EDX analyses.

4. On the UV spectrum the authors claim that the 290 nm band observed is due to the Cu-O bond. Do the authors mean that this is a n-pi* transition from the oxygen n orbital?

Authors: The representative bands of CuO in the 250–350 and 600–800 nm regions are attributed to Cu2+–O charge-transfer transitions and d–d transitions of dispersed CuO species, respectively.

5. On page 5, author make the following statement: “… and decreasing the catalyst loading is not good for CuAAC reactions”. Such statement should be supported with a reference to literature.

Authors: This statement was reported based on the obtained results in Table 1, see page 5.

6. Later on the same paragraph the authors refer to Table 1 entries 16-21. Such entries do not exit in the table. The last entry is number 12. Please address this.

Authors: The entries 16-21 were modified and the correct entries appaear now as entries 10-12, see page 5.

7. The authors claim that they always obtain >90% isolated yields. It is surprising to see that they do not observe any alkyne homocoupling reaction, given that the alkyne is the limiting reagent. Do the authors have any explanation for this observation?

Authors: To remove the product of the homocoupling reaction, we have used alkynes as a limiting reagent.

8. The authors claim in table 2 that the catalyst loading is 3 mol% Cu. To calculate that 3% loading did they use the 0.7% content or did they use the 1.87%? This also has implications later on on Table 3 about the reusability of the catalyst. Given that the copper content after 3 cycles is only 0.02% weight by EDX, this implies that more than 99% of the copper present in the catalyst has been lost after 3 cycles. This presents a clear barrier to the use of this catalyst, since one of the main advantages of heterogeneous catalysts and being able to use them in continues processing, instead of batch chemistry, is rapidly degraded.

Authors: We thank the reviewer for this excellent remark. We belive that this drop of the copper content after the 3th cycle is most probably due to the leaching of copper caused by the instability of the catalytically active copper(I) species in water and under air atmosphere after continuous processing during prolonged reaction times. This issue is under investigation to improve the work of Cu(II)-CA under sustainable real industrial conditions.

9. What are the reaction conditions for the reusability in Table 3? Are they the same as Table 2? Can the author please include the conditions in the table caption as they did in table 2 please?

Authors: We thank the reviewer for this remark. The reaction conditions used in the reusability study were mentioned at the caption of Table 3, see page 7.

Round 2

Reviewer 2 Report

The authors have revised the manuscript taking into consideration the comments of the reviewers. The article can be published in this journal after changes made:

1. This reviewer not satisfied with the answers on the questions â„–2, â„–4 (I would not illustrate that the Cu ions bonded covalently on Scheme 1. Not enough data to support this fact).

2. Instead of “azdides” should be “azides” (page 2, line 50).

3. In the sentence (page 10, lines 248-259) after the word “sustainable” probably have to be the word “for” … CuAAC reactions…

Minor editing of English language required.

Author Response

Point-by-point responses to the reviewers:

Reviewer #2:

The authors have revised the manuscript taking into consideration the comments of the reviewers. The article can be published in this journal after changes made:

1. This reviewer not satisfied with the answers on the questions â„–2, â„–4 (I would not illustrate that the Cu ions bonded covalently on Scheme 1. Not enough data to support this fact).

Authors: Scheme 1 was modified as suggested by the reviewer, pointing to the occurrence of weak interactions between copper(II) ions with the cellulose backbone rather than coordinate covalent bonding.

2. Instead of “azdides” should be “azides” (page 2, line 50).

Authors: Spelling and style were checked throughout the main text of the manuscript.

3. In the sentence (page 10, lines 248-259) after the word “sustainable” probably have to be the word “for” … CuAAC reactions…

Authors: The mentioned sentence was revised, see page 10.

Reviewer 3 Report

Point-by-point responses to the Authors:

Reviewer #3:

This manuscript reports the development of new catalyst for copper click chemistry based on cellulose. The catalyst proposed works well in water and authors claim it to be heterogenous. The cellulose materials are characterized with a series of techniques: EDX, IR, SEM and ICP. Results are clearly presented, but some of the results seem to be in contradiction.

This paper might be of interest to researchers interested in green chemistry and copper catalyzed C-C bond forming reactions.

A main concern of the manuscript is that the authors claim that the catalyst is heterogeneous, however the experiment to prove this, “hot filtration”, seems to provide results contradicting other results. For instance, the catalyst quickly degrades after a few cycles (60% yield after 3 cycles) and the content of copper as determined by EDX decreases from 1.87% to 0.02%. Such a decrease indicates that the cellulose-copper bonds are labile and hence a lot of the copper must be in solution during the catalysis. This seems to contradict the results from the “hot filtration” test. Nevertheless, this is an interesting study, and the manuscript is well written and presented. This reviewer believes that the readers of Int J Mol Sci will enjoy the content of the manuscript and considers it suitable for publication after the following points have been addressed:

1. On page two, can the authors add the carbon labelling scheme to the Scheme 1, please? The text describing the IR refers to C2, C3 and C6 and those do not appear in the scheme.

Authors: Following the reviewer's suggestion, carbon numbering was added in Scheme 1.

Reviewer #3:  Addressed.

2. While describing changes in the IR, the authors describe that no new bands appear when comparing the IR of cellulose to the cellulose-Cu catalyst. This is not surprising, giving that the copper content of the catalyst is only 0.7%. It is surprising that they observe such a difference in the intensity of the bands, and it begs to wonder if the difference arises from a lack of an internal reference.

Authors: We thank the reviewer for drawing our attention to this remark. The internal reference was used in the FTIR analysis, but no new bands were found due to the low loading of copper ions in the Cu(II)-CA catalyst.

Reviewer #3:  Addressed.

3. The authors claim that the copper content by EDX is 1.87% wt, while the copper content by ICP is 0.7% wt. This is a large discrepancy that can lead to extremely large differences in catalyst loading. One simple question is if the two samples measured by the different techniques were obtained from different batches. Indicating that the copper absorption by the cellulose is hugely variable. The authors explain the difference between the two values by the different techniques saying that the EDX only gives surface content, if this was correct, then the surface content by EDX should be lower than the total content obtained by ICP. This result is hugely contradictory and it must be addressed prior to publication.

Authors: The ICP analysis was performed using 100 mg of prepared catalyst while the EDX analysis was performed using only 10 mg of this catalyst. These findings indicate that the large copper content is located in the CA surface, which can explain the high difference values between ICP and EDX analyses.

Reviewer #3:  The authors answer still does not address the main contradiction of these results. ICP is a technique that gives total content of elements, if it is well calibrated. Therefore it does not make sense that the Copper content by ICP is lower than by EDX. The authors answer that the EDX shows less content because it only shows the copper in the surface contradicts the results reported. EDX gives them 1.87%, which is higher than 0.7%, not lower as the authors claim. Did the authors run a calibration curve for the ICP? What is the precision of their measurement?

4. On the UV spectrum the authors claim that the 290 nm band observed is due to the Cu-O bond. Do the authors mean that this is a n-pi* transition from the oxygen n orbital?

Authors: The representative bands of CuO in the 250–350 and 600–800 nm regions are attributed to Cu2+–O charge-transfer transitions and d–d transitions of dispersed CuO species, respectively.

Reviewer #3:  Addressed.

5. On page 5, author make the following statement: “… and decreasing the catalyst loading is not good for CuAAC reactions”. Such statement should be supported with a reference to literature.

Authors: This statement was reported based on the obtained results in Table 1, see page 5.

Reviewer #3:  Addressed.

6. Later on the same paragraph the authors refer to Table 1 entries 16-21. Such entries do not exit in the table. The last entry is number 12. Please address this.

Authors: The entries 16-21 were modified and the correct entries appaear now as entries 10-12, see page 5.

Reviewer #3:  Addressed.

7. The authors claim that they always obtain >90% isolated yields. It is surprising to see that they do not observe any alkyne homocoupling reaction, given that the alkyne is the limiting reagent. Do the authors have any explanation for this observation?

Authors: To remove the product of the homocoupling reaction, we have used alkynes as a limiting reagent.

Reviewer #3:  Addressed.

8. The authors claim in table 2 that the catalyst loading is 3 mol% Cu. To calculate that 3% loading did they use the 0.7% content or did they use the 1.87%? This also has implications later on on Table 3 about the reusability of the catalyst. Given that the copper content after 3 cycles is only 0.02% weight by EDX, this implies that more than 99% of the copper present in the catalyst has been lost after 3 cycles. This presents a clear barrier to the use of this catalyst, since one of the main advantages of heterogeneous catalysts and being able to use them in continues processing, instead of batch chemistry, is rapidly degraded.

Authors: We thank the reviewer for this excellent remark. We belive that this drop of the copper content after the 3th cycle is most probably due to the leaching of copper caused by the instability of the catalytically active copper(I) species in water and under air atmosphere after continuous processing during prolonged reaction times. This issue is under investigation to improve the work of Cu(II)-CA under sustainable real industrial conditions.

Reviewer #3:  Addressed.

9. What are the reaction conditions for the reusability in Table 3? Are they the same as Table 2? Can the author please include the conditions in the table caption as they did in table 2 please?

Authors: We thank the reviewer for this remark. The reaction conditions used in the reusability study were mentioned at the caption of Table 3, see page 7.

Reviewer #3:  Addressed.

Author Response

Point-by-point responses to the reviewers:

The authors answer still does not address the main contradiction of these results. ICP is a technique that gives total content of elements, if it is well calibrated. Therefore it does not make sense that the Copper content by ICP is lower than by EDX. The authors answer that the EDX shows less content because it only shows the copper in the surface contradicts the results reported. EDX gives them 1.87%, which is higher than 0.7%, not lower as the authors claim. Did the authors run a calibration curve for the ICP? What is the precision of their measurement?

Authors: We thank the reviewer for drawing our attention to this remark. The ICP analyses were revised and the correct loading of copper ions is 1.37%. Consequently, the loading of copper in the CuAAC reactions was corrected.

Hoping that this second revised version could be finally accepted for publication in the International Journal of Molecular Sciences, and thanking you for your time and attention.